# The Caliber of Segmental and Subsegmental Vessels in COVID-19 Pneumonia Is Enlarged: A Distinctive Feature in Comparison with Other Forms of Inflammatory and Thromboembolic Diseases

**DOI:** 10.3390/jpm12091465

**Published:** 2022-09-07

**Authors:** Maria-Chiara Ambrosetti, Giulia Battocchio, Stefania Montemezzi, Filippo Cattazzo, Tissjana Bejko, Evelina Tacconelli, Pietro Minuz, Ernesto Crisafulli, Cristiano Fava, Giancarlo Mansueto

**Affiliations:** 1Institute of Radiology, Department of Diagnostics and Public Health, Policlinico GB Rossi, University of Verona, 37129 Verona, Italy; 2Radiology Unit, Department of Pathology and Diagnostics, Azienda Ospedaliera Universitaria Integrata-Verona, 37129 Verona, Italy; 3General Medicine & Hypertension Unit, Department of Medicine, University of Verona, 37129 Verona, Italy; 4Infectious Disease Unit, Department of Diagnostics and Public Health, University of Verona, 37129 Verona, Italy; 5Respiratory Diseases Unit, Department of Medicine, University of Verona, 37129 Verona, Italy

**Keywords:** COVID-19 virus disease, CT scan, SARS-CoV, pulmonary vessel dilatation

## Abstract

Background: The purpose of this study was to compare COVID-19 patients’ vessel caliber with that of normal lungs and lungs affected by other inflammatory and thromboembolic processes. Methods: between March and April 2020, 42 patients affected by COVID-19 pneumonia (COV-P) underwent CT scans of the lungs at Verona University Hospital for clinical indications. The lung images of four different groups of patients were compared (normal lung (NL), distal thromboembolism (DTE), and bacterial and fungal pneumonia (Bact-P, Fung-P)) by a radiologist with four years of experience. Results: The COV-P patients’ segmental and subsegmental vessels, evaluated as the ratio with the corresponding bronchial branch (V/B ratio), were larger, with respect to the NL the DTE groups, in the apparently healthy parenchyma, a result confirmed in the zones of opacification with respect to the Bact-P and Fung-P groups. Conclusions: This was the first study to show, by comparative analysis, that COVID-19 patients’ segmental and subsegmental vessel calibers are significantly enlarged. This is a distinctive feature of COVID-19 pneumonia, suggesting its distinct pathophysiology as compared to other inflammatory and thromboembolic diseases and alerting radiologists to consider it when evaluating the CT scans of suspected patients.

## 1. Introduction

Since the outbreak of the SARS-CoV-2 infection in the Hubei province in China in late December 2019 [1], it has been evident that a prevalent target of coronavirus disease-19 (COVID-19) is the lungs, with a broad spectrum of clinical respiratory syndromes [2]. Clinical features range from either an asymptomatic presentation or mild upper airway symptoms to progressive and life-threatening respiratory distress [3]. A recent study has suggested that microvascular thrombotic processes, together with ARDS, may play a role in respiratory failure in COVID-19 patients [4]. Similar to the clinical manifestations, a wide range of radiologic patterns have been described, from sole ground-glass opacities (GGO) to consolidative pulmonary opacities, often with a bilateral and peripheral lung distribution [5]. Therefore, the presence of GGO, mainly in patients with mild forms of COVID-19, with symptoms such as interlobular septal thickening, parenchymal consolidation of the air bronchogram, pleural effusion, and pulmonary fibrosis, which is more prevalent in the severe clinical form, has characterized the chest CT features of COVID-19 pneumonia [6]. Recently, great efforts were made to introduce deep learning techniques for the CT chest analysis of patients with COVID-19 pneumonia in order to improve the diagnostic accuracy, together with the identification of a prognostic score [7,8,9,10,11]. Changes in pulmonary vascularization, such as segmental or subsegmental vascular enlargement, have been also described as a specific feature of the COVID-19 disease [12,13]. To date, however, the cause of the vessel enlargement has not been clarified. Some hypotheses have been proposed, which are not mutually exclusive [14]. In particular, microvascular pulmonary thrombi and emboli have often been associated with COVID-19 pneumonia and are also related to the severe hypoxemia that characterizes the disease [15]. Another possibility is the profound inflammation and the vasodilatory effects of cytokines that are released in abundance in response to the pulmonary parenchymal damage itself, which also correlate with the so-called “cytokine storm” [16]. A final hypothesis is related to the induction of endothelitis and angiogenesis, mediated by the SARS-CoV-2 infection [17]. 

To our best knowledge, there are no published studies focusing on the caliber of the segmental and subsegmental branches of the pulmonary arteries in patients with COVID-19 pneumonia. In order to clarify whether the enlargement of the caliber of the vessels is a COVID-19-specific phenomenon or whether it is present in other primary thromboembolic or inflammatory conditions, we selected different control groups: (i) patients without evidence of lung disease (normal lung (NL)); (ii) patients with micro-vessel embolisms (distal thromboembolism (DTE)); and patients with distinct forms of pneumonia of, respectively, (iii) bacterial (Bact-P) and (iv) fungal origins (Fung-P).

## 2. Methods

### 2.1. Study Population

All patients admitted to the emergency department or COVID units of our hospital from 1 March to 25 April 2020, who had respiratory symptoms, were positive for SARS-CoV-2, as confirmed by a reverse-transcription polymerase chain reaction (RT-PCR) test, and who underwent a chest CT, were enrolled in this retrospective study. The exclusion criteria were the presence of mild-severe respiratory artifacts on the chest CT.

We also collected CT images from four control groups: patients who underwent chest CT before January 2020 or after April 2020, without evidence of lung disease or COVID-19 positivity (NL); patients with pulmonary embolisms involving only the distal branches of the pulmonary arteries (DTE); and patients with confirmed bacterial (Bact-P) or fungal (Fung-P) pneumonia, with the pathogen isolated upon blood culture or bronchiolar alveolar lavage (BAL), or in the case of fungal infection, the serum positivity of beta-glucan. Groups DTE, Bact-P, and Fung-P included only patients examined before December 2019, who underwent a CT exam for respiratory symptoms. None of the examined patients had a history of bronchiectasis or evidence of bronchiectasis based on previous CT scans (in particular, we observed CT scans available for six of the patients and plain radiography images for eight of them, and none showed signs compatible with bronchiectasis).

The prospective and retrospective study was approved by our local institutional review board (IRB; 2695CESC), and written informed consent was obtained from all study participants. The COVID-19 patients were prospectively enrolled in a registry (COVID-19 2577CESC) and gave oral informed consent, allowing the use of their data for the research.

### 2.2. Image Acquisition

All chest scans were performed on a 64- or 6-row multiple detector computed tomography scanner (Brilliance 64 or 6, Philips, Amsterdam, the Netherlands), with the patients in the supine position, with the arms extended above the head, at end-inspiration. 

For the patients in the Bact-P, Fung-P, and COV-P groups, without suspicions of pulmonary embolism, the acquisition was performed without contrast administration.

For the other patients, the acquisition was obtained after a weight-based amount (1.4 mL/Kg) of high-concentration contrast agent (370 mgI/mL, Ultravist 370, BayerScheringPharma, Berlin, Germany) was administered at a rate of 3–4 mL/s, followed by a 50 mL saline bolus administered by means of a dual-head injector (Medrad Stellant, Indianola, IA, USA) through an antecubital vein. The scanning time was determined using a bolus-tracking technique, with the monitoring region of interest in the thoracic aorta for the NL group, or by performing computerized tomography pulmonary angiography (CTPA) on the pulmonary trunk of patients with suspected pulmonary embolisms. The scan delay after trigger was 15 s for the NL patients and the minimum for the CTPA. 

All scans were reconstructed with a sharp lung filter at a slice thickness of 1.25 mm.

### 2.3. Image Analysis

One radiologist with 4 years of experience in chest imaging analyzed the cases using a PACS workstation for axial images with a parenchymal window. 

In all patients, the radiologist measured the caliber of one segmental branch of both the right and left pulmonary arteries, their corresponding bronchial branch, one subsegmental branch of right and left pulmonary arteries, and their corresponding bronchial branch in the healthy lung parenchyma, without lung opacification.

For patients in the COV-P, Bact-P, and Fung-P groups, the radiologist also measured the caliber of one segmental and one subsegmental branch of the pulmonary artery, together with their corresponding bronchial branches in the main area of lung opacification. To standardize the results, the radiologists measured the diameter only using the cross-sectional images of the segmental arterial branches 2 cm downstream of their onset and of the subsegmental branches at the level of the 7–8th generation. The adjacent bronchial branch diameter was measured, respectively, at the same level (Figure 1). To reduce inter-patient constitutional variability, the measurements of the healthy lung parenchyma were performed on the lower lobes.

To normalize the measures, the ratio between the caliber of each vessel and its corresponding bronchus (V/B RATIO) was calculated for each measurement site.

In COV-P patients, the level of C-reactive proteins, PaO_2_/FiO_2_, and oxygen saturation measured at the emergency department were collected.

### 2.4. Statistics

The calibers of the segmental and subsegmental pulmonary artery branches and their corresponding bronchial branches, as well as the V/B ratios, were compared across groups using the Kruskal–Wallis test and a post hoc Mann–Whitney U test, implemented using the “Statistical Package for Social Sciences” software (SPSS/PC for Mac, version 26.0, IBM Corporation, Chicago, IL, USA). The graphs were designed using GraphPad Prism software (version 7.00, GraphPad Software, La Jolla, CA, USA, www.graphpad.com, accessed on 1 September 2022), and *p* values < 0.05 were considered statistically significant. 

## 3. Results

Between the 1st of March and the 25 April 2020, 56 patients with respiratory symptoms and who were positive for SARS-CoV-2, as confirmed by RT-PCR, underwent CT at our institute. Fourteen patients underwent CTPA who had suspected pulmonary embolisms, and the other 42 patients underwent non-enhanced chest CT. Fourteen patients were excluded due to mild-severe motion artifacts identified from the CT, with the other 42 patients forming the COV-P group (12 females and 30 males, mean age 64.5 ± 13.5 years). Most of them had moderate to critical COVID-19 pneumonia, as demonstrated also by their average PaO_2_/FiO_2_ ratio of 241 ± 117 (available for 35 patients upon admission). Only 12 of the 42 patients had a blood oxygen saturation of >94% while breathing in ambient air.

The NL group (18 females and 24 males, mean ± SD age 65.3 ± 14.5 years) included 42 patients without respiratory symptoms, who underwent CT at our Institute for follow-up testing for pathologies not involving the lungs. A total of 30 patients underwent the CT exam before December 2019, and 12 patients between the 4th and the 7th of May. 

The DTE group (18 females and 24 males, mean age 66.0 ± 17.1 years) included 42 patients who underwent CTPA with suspected pulmonary embolisms, with the evidence of the endoluminal defects not involving the main pulmonary arteries but only their distal branches. 

The Bact-P group (14 females and 28 males, mean age 59.8 ± 13.7 years) included 42 patients who underwent CT of the chest for pulmonary infections, and from whom a bacterium was isolated through blood culture or BAL. 

The Fung-P group (9 females and 33 males, mean age 57.0 ± 14.7 years) included 42 patients who underwent CT of the chest with suspected pulmonary infections, from whom a fungus was isolated through blood culture or BAL, or by the demonstration of beta-glucan positivity.

Age, but not sex, was slightly different between only two of the groups (*p* < 0.027 for Fung-P vs. DTE). However, in the univariate regression analysis, neither the V/B ratio of the segmental or subsegmental branches nor the V/B ratio in the area of lung opacification were significantly associated with age.

The V/B ratio of the segmental branches of the healthy lung parenchyma in the COV-P group was significantly higher than those of all the other groups (*p* < 0.0001; Table 1, Figure 2 and Figure 3). The V/B ratio of the subsegmental branches of the healthy lung parenchyma in the COV-P group was the highest among the groups and significantly higher than those of the NL, DTE, and Fung-P groups (*p* ≤ 0.0001; Table 1, Figure 4).

When analyzing the measurements of the segmental V/B ratios in patients in the COV-P, Bact-P, and Fung-P groups in the area of lung opacification, the values of group COV-P were always significantly higher than the corresponding values of the two other groups (Table 2, Figure 3 and Figure 5).

The V/B ratio values of the subsegmental branches in the area of lung opacification in the COV-P group were the highest among the groups and significantly higher than that of the Fung-P group (*p* = 0.0003; Table 2, Figure 6).

Among the groups of patients with pulmonary infections (COV-P, Bact-P, and Fung-P), considering the calibers of the segmental and subsegmental branches of the pulmonary arteries in the area of lung opacification, the calibers in group COV-P were significantly higher than those in the Bact-P and Fung-P groups. No significant difference was found in the segmental bronchi calibers, while the caliber of the subsegmental bronchi was significatively higher group COV-P than in groups Bact-P and Fung-P (*p* < 0.0001; Table 2).

The level of the C-reactive proteins, available in 31 COV-P patients, was higher in patients with a V/B ratio above the median as compared to those whose ratio was below the median (124 ± 93 vs. 62 ± 49 mg/L; *p* < 0.05). Although this is not a prognostic study, neither the PaO_2_/FiO_2_ ratio nor the ROX index (measured at the emergency department upon admission and upon the patients’ clinical deterioration) were associated with the V/B ratios of segmental or subsegmental branches nor the V/B ratio in the area of lung opacification.

## 4. Discussion

Severe COVID-19 disease manifests as a devastating pneumonia characterized by diffuse ground-glass and consolidative pulmonary opacities, often with a bilateral and peripheral lung distribution, leading to respiratory failure. The low blood oxygenation is not only driven by the extensive involvement of the lung or the development of the acute respiratory distress syndrome (ARDS), but also by microvascular thrombotic processes, contributing to the high mortality rates of this disease [18].

In the present study, we focused our attention on the caliber of the segmental and subsegmental branches of the pulmonary arteries in the healthy lung parenchyma, in correspondence with the lung opacities. Not surprisingly, the caliber of the COVID-19 patients’ segmental and subsegmental vessels was, on average, significantly higher with respect to the one of patients, who was affected by other forms of infectious or thromboembolic diseases. These enlargements were present both in sites of radiologically active disease (opacifications) and in apparently healthy parenchyma. Thus, our study proves that this is a feature that is especially specific to COVID-19 pneumonia and is more pronounced than in other inflammatory or thromboembolic disease. 

Other studies have already reported the same finding, but without any comparison between groups. Caruso et al. described enlarged subsegmental pulmonary vessels in 52 out of 58 COVID-19 patients (89%), defining vessel enlargement as a vascular caliber of >3 mm. Vessel enlargement in proximity to areas with GGO was described, and the authors postulated that this could be related to thrombo-inflammatory processes [12]. Parry et al. even observed segmental and subsegmental pulmonary vascular enlargements on chest CT images [14]. At variance with these studies, in order to normalize the caliber of the measured vessels, which may be influenced by the physical features of individual patients, and to reduce the variability due to the age of the patients, we decided to measure the corresponding bronchial branches and to calculate the ratio between each arterial and bronchial branch. The ratio between the segmental branch of the pulmonary artery and the corresponding bronchus has already been proposed, together with other imaging features, as a method for identifying the presence of pulmonary hypertension non-peripheral pulmonary vascular disorders at CT [19].

Moreover, not only did we compare the calibers of subjects without evidence of lung disease, but we also included patients with pulmonary embolism of the segmental or subsegmental branches of the pulmonary arteries and with other infections. In particular, with respect to the patients with bacterial and fungal pneumonia, the V/B segmental ratio of the lung opacifications was significantly higher among the COVID-19 patients and, since the calibers of the correspondent bronchial branches among the groups were not significatively different, this was due to the dilatation of the segmental vessels in the COVID-19 patients compared to the other groups (Table 2). At the same time, the V/B subsegmental ratio of the lung opacifications was higher in the COVID-19 patients but not significant because, together with the dilatation of the subsegmental vessels, a dilatation of the correspondent bronchial also occurred, with a consequently lower increase in the V/B ratio (Table 2). The dilatation of the distal bronchial branches of the lung opacifications, such as the presence of bronchiectasis, has already been described in COVID-19 patients and could be related to the evolution of a fibrosing lung pattern [20].

The pathophysiology of such an enlargement of the lung small vessels is poorly understood but may derive from the complex interplay of all the mechanisms that were previously proposed: thromboembolism, inflammation, and endothelial impairment [12]. Indeed, it could reflect the presence of micro-thrombi in the small pulmonary vessels. McGonagle et al. suggested that hypoxemia might determine endothelial dysfunction and activate the coagulation cascade, and it might also play a role in adjacent small pulmonary vascular thrombosis, while other factors, including mechanical ventilation, might contribute [21]. In any case, it is worth underlining that none of the COVID-19 patients included in the present study were intubated at the time of the CT scan. 

Several studies have observed high plasma levels of the proinflammatory cytokines in COVID-19 patients admitted to intensive care units, known as the so-called “cytokine storm.” These pro-inflammatory cytokines might trigger the coagulation system [16]. Other authors have suggested that vascular enlargement may be due to pro-inflammatory factors that determine vascular hyperemia [22]. In a recent autopsy study, Ackermann et al. examined 7 lungs obtained from patients who died from COVID-19 and compared them with 7 lungs obtained from patients who died from acute respiratory distress syndrome (ARDS), secondary to influenza A(H1N1) infection, and 10 age-matched, uninfected control lungs. The lungs of patients with COVID-19 showed distinctive vascular features, consisting of severe endothelial injury associated with the presence of intracellular virus and disrupted cell membranes. Histologic analysis of the pulmonary vessels in patients with COVID-19 showed widespread thrombosis, with microangiopathy. Furthermore, vascular angiogenesis was observed to distinguish the pulmonary pathobiology of COVID-19 [23]. As the V/B ratio of the segmental and subsegmental branches was higher in all the groups of patients with pulmonary infections as compared to those with pulmonary embolisms, we can suggest that the role of inflammation could be predominant in the small-vessel enlargement. To add to the inflammatory hypothesis, when comparing the level of C-reactive proteins measured at the emergency department in 31 COVID-19 patients, divided according to the median value of the segmental vessel V/B ratio (1.2), we noticed higher values in patients with a V/B ratio above the median as compared to those with a value below the median (124 ± 93 vs. 62 ± 49 mg/L; *p* < 0.05).

The limitations of our study include the facts that its design was retrospective, only patients with clinical indications of the CT were evaluated, and the ages between groups were slightly different due to the pandemic situation. Only one radiologist, due to limited time, who was not blinded to the lung pathologies, analyzed the images. Either CT or CTPA images were evaluated, and interstitial forms of pneumonia were missing, as were the clinical and laboratory data of many patients, so that a completely trustworthy comparison between the groups and subgroups was not feasible. However, as far as we know, this was the first study focusing on the quantitative evaluation of vessel enlargements in COVID-19 patients, comparing this group of patients with different control groups, and it has discovered important information about the specific lung involvement in this disease.

In conclusion, we have proven that the caliber of the segmental and subsegmental vessels in COVID-19 pneumonia is significantly higher than that of normal lungs and in other forms of TE and infectious diseases, at least in the sites of parenchymal opacification. The process leading to this vessel enlargement could be the sum of the profound inflammation and distal thromboembolic processes that, together, contribute to the pathophysiology of this devastating disease. Radiologists should pay attention to this characteristic when analyzing CT scans of patients with suspected COVID-19 pneumonia. 

## Figures and Tables

**Figure 1 jpm-12-01465-f001:**
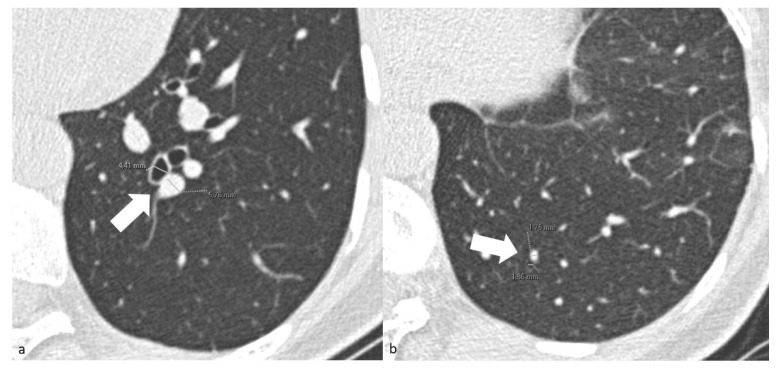
(**a**) Chest axial CT images of a COV-P patient. Measurements of the segmental arterial (4.41 mm) and adjacent bronchial branches (5.76 mm) (arrow). (**b**) Chest axial CT images of a COV-P patient. Measurements of the and subsegmental arterial (1.56 mm) and adjacent bronchial branches (1.75 mm) (arrow) of healthy lung parenchyma of the lower left lobe.

**Figure 2 jpm-12-01465-f002:**
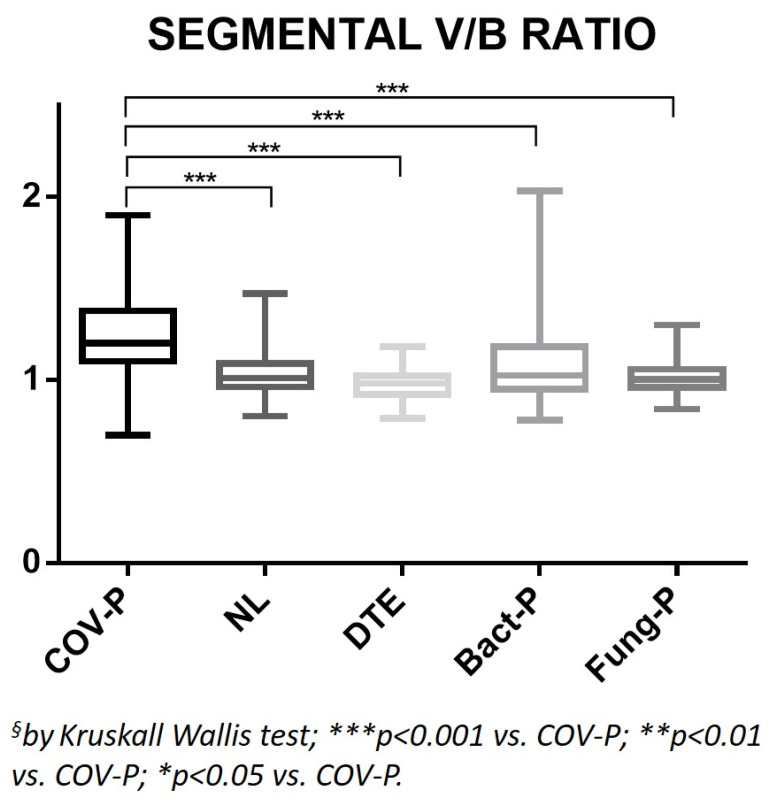
Ratio between the caliber of the segmental branches of the pulmonary arteries and their corresponding bronchial branches, measured on healthy lung parenchyma. COV-P, COVID-19 pneumonia; NL, normal lung; DTE, distal thromboembolism; Bact-P, bacterial pneumonia, Fung-P, fungal pneumonia.

**Figure 3 jpm-12-01465-f003:**
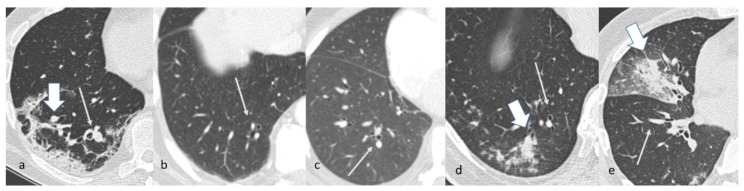
Chest CT images of patients in, respectively, group COV-P (**a**), NL (**b**), DTE (**c**), Bact-P (**d**), and Fung-P (**e**). Note the segmental branches of the pulmonary artery and the corresponding bronchus on the healthy parenchyma (thin arrows (**a**–**e**)) and the area of lung opacification (wide arrows (**a**,**d**,**e**)).

**Figure 4 jpm-12-01465-f004:**
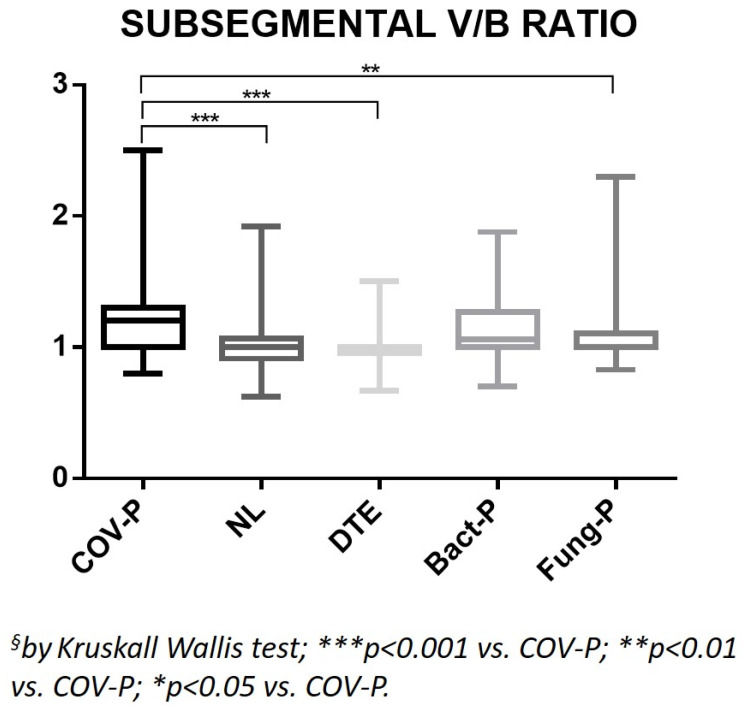
Ratio between the caliber of the subsegmental branches of the pulmonary arteries and their corresponding bronchial branches, measured on healthy lung parenchyma. COV-P, COVID-19 pneumonia; NL, normal lung; DTE, distal thromboembolism; Bact-P, bacterial pneumonia, Fung-P, fungal pneumonia.

**Figure 5 jpm-12-01465-f005:**
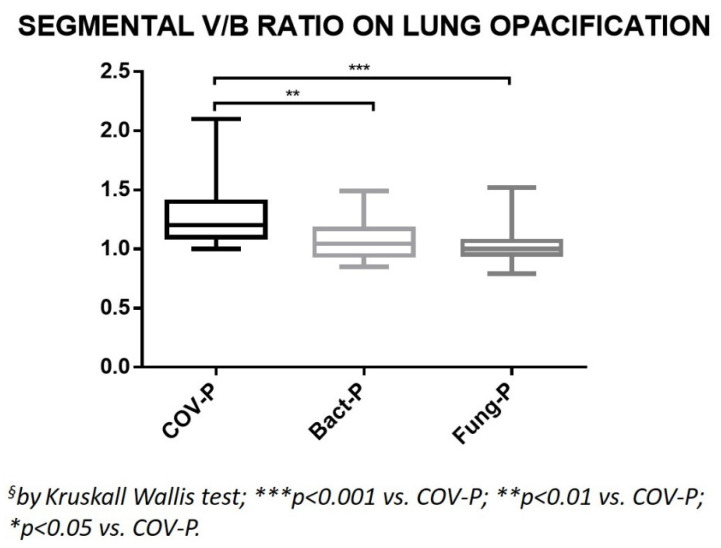
Ratio between the caliber of the segmental branches of the pulmonary arteries and their corresponding bronchial branches, measured on the area of lung opacification. COV-P, COVID-19 pneumonia; Bact-P, bacterial pneumonia; Fung-P, fungal pneumonia.

**Figure 6 jpm-12-01465-f006:**
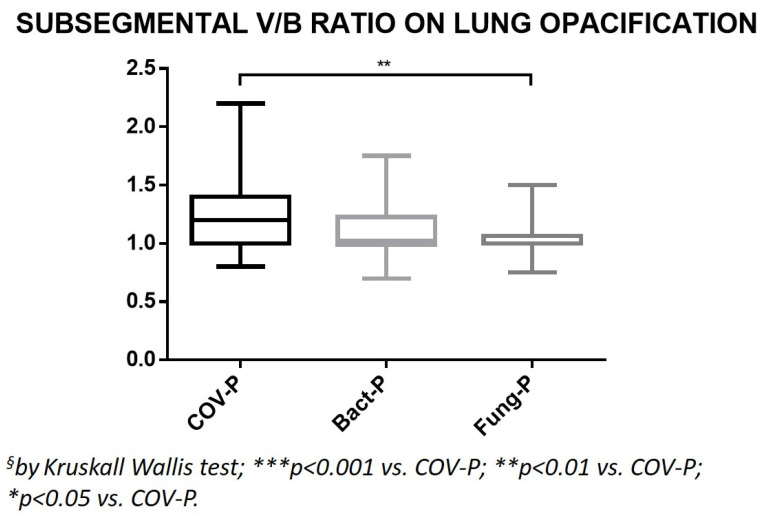
Ratio between the caliber of the subsegmental branches of the pulmonary arteries and their corresponding bronchial branches, measured on lung opacification. COV-P, COVID-19 pneumonia; Bact-P, bacterial pneumonia; Fung-P, fungal pneumonia.

**Table 1 jpm-12-01465-t001:** Ratio between the caliber of the segmental and subsegmental branches of the pulmonary arteries and their corresponding bronchial branches, measured on healthy lung parenchyma.

	COV-P	NL	DTE	Bact-P	Fung-P	*p* ^§^
**Segmental V/B RATIO**	1.23 ± 0.24	1.03 ± 0.13 ***	0.97 ± 0.07 ***	1.12 ± 0.26 ***	1.02 ± 0.10 ***	<0.0001
**Subsegmental V/B RATIO**	1.22 ± 0.27	1.01 ± 0.22 ***	1.02 ± 0.15 ***	1.17 ± 0.27	1.07 ± 0.21 ***	<0.0001

COV-P, COVID-19 pneumonia; NL, normal lung; DTE, distal thromboembolism; Bact-P, bacterial pneumonia, Fung-P, fungal pneumonia. ^§^ by Kruskal–Wallis test; *** *p* < 0.001 vs. COV-P; ** *p* < 0.01 vs. COV-P; * *p* < 0.05 vs. COV-P.

**Table 2 jpm-12-01465-t002:** Calibers of the segmental and subsegmental branches of the pulmonary arteries and their corresponding bronchial branches, measured in the areas of lung opacification, and the V/B ratios calculated.

	COV-P	Bact-P	Fung-P	*p* ^§^
**Segmental V/B ratio**	1.28 ± 0.28	1.09 ± 0.18 **	1.03 ± 0.14 ***	<0.0001
**Subsegmental V/B ratio**	1.23 ± 0.29	1.11 ± 0.24	1.02 ± 0.13 **	0.0005
**Segmental vessel caliber (mm)**	5.83 ± 1.19	4.73 ± 1.06 ***	4.92 ± 0.62 **	<0.0001
**Segmental bronchial caliber (mm)**	4.66 ± 0.99	4.41 ± 0.95	4.84 ± 0.77	n.s.
**Subsegmental vessel caliber (mm)**	2.97 ± 0.85	1.99 ± 0.57 ***	1.78 ± 0.33 ***	<0.0001
**Subsegmental bronchial caliber (mm)**	2.44 ± 0.60	1.80 ± 0.39 ***	1.77 ± 0.37 ***	<0.0001

COV-P, COVID-19 pneumonia; Bact-P, bacterial pneumonia, Fung-P, fungal pneumonia. ^§^ by the Kruskal–Wallis test; *** *p* < 0.001 vs. COV-P; ** *p* < 0.01 vs. COV-P; * *p* < 0.05 vs. COV-P.

## Data Availability

Clinical and imaging data, after obtaining informed consent from all the patients, were collected using our PACS workstation and clinical diary.

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
