# Peer review of "The Caliber of Segmental and Subsegmental Vessels in COVID-19 Pneumonia Is Enlarged: A Distinctive Feature in Comparison with Other Forms of Inflammatory and Thromboembolic Diseases"

_jpm, 2022, doi:10.3390/jpm12091465_

Round 1
Reviewer 1 Report
Although this study is interesting, there are some issues that should be reconsidered.
How were cases with pre-existing bronchiectasis and cardiovascular disease prior to COVID19? Or did you not consider the issue?
The diameter of pulmonary artery change with age (AJR Am J Roentgenol. 2003 Feb;180(2):513-8. ). It has also been reported that the bronchus shows large fluctuations depending on the measurement site (Radiology. 2005 Feb;234(2):595-603. ). Under such circumstances, it is unclear whether the V/B ratio has sufficient reliability.
Author Response
Q1 How were cases with pre-existing bronchiectasis and cardiovascular disease prior to COVID19? Or did you not consider the issue?
REV1 Q1. We thank the reviewer for the comment. None of the examined patients had a history of bronchiectasis or evidence of them at CT (please see page 6, lines 118-120). Regarding cardiovascular disease, we have data only in COVID-19 patients. Of these, 20/42 (47.6%) had a history of hypertension and 4/42 (9.5%) had a history of prior episodes of heart failure or atrial fibrillation whereas 7/42 (16.7%) patients had diabetes.
Q2A The diameter of pulmonary artery change with age (AJR Am J Roentgenol. 2003 Feb;180(2):513-8).
Q2B It has also been reported that the bronchus shows large fluctuations depending on the measurement site (Radiology. 2005 Feb;234(2):595-603. ). Under such circumstances, it is unclear whether the V/B ratio has sufficient reliability.
REV1 Q2A: We thank the reviewer for the observation. As he correctly reported, it is well-known that the caliber of pulmonary arteries and bronchi changes with ageing, both increasing. We tried to overcome this problem not considering the mere caliber of the pulmonary arteries but the ratio between the caliber of the pulmonary artery and the corresponding bronchial branchi. By comparing the ages in different groups of patients by the One-Way ANOVA test we found that age was statistically different in two groups (Fung-P vs DTE). Anyhow, in univariate regression analysis, neither the V/B ratio of the segmental or subsegmental branches nor the V/B ratio in the area of lung opacification were significantly associated with age Thus, we think that our results remain trustable. We have added a sentence in the results section (please see page 9, lines 192-195)
REV1 Q2B: As the reviewer pointed out, the caliber of the bronchi shows large fluctuations depending on the measurement site. For this reason, and as already done in previous studies, we standardized the measurement of the segmental and subsegmental bronchial branches respectively 2 centimeters downstream their onset and at the level of the 7-8th generation (please see page 7, lines 151-156 and page 11, lines 244-245).
Reviewer 2 Report
This is an interesting study since it addresses the quantification of specific biomarkers related to the increased caliber of segmental and subsegmental vessels in 2 COVID-19 pneumonia.
The authors should address the following issues before resubmitting:
a) Why did the authors did not use existing CT data from normal patients to compare with the COVID-19 caliber.
b) Are there any common characteristics/features within the COVID-19 patients who died that are related to the increases in caliber (e.g. SOFA, APACHE, oxygenation, crackles, sqwuaks, etc).
c) There are a lot of studies analysing CTs with deep learning techniques. The authors are urged to review the literature from the past 2 years for any studies that are directly or indirectly linked to the current paper.
d) ARDS is very much related with COVID-19, but it is also present in other lung diseases such as COPD, asthma etc. How can ARDS be predicted from the proposed biomarker?
Author Response
Q1. Why did the authors did not use existing CT data from normal patients to compare with the COVID-19 caliber.
REV2 Q1: Even though there are plenty of existing CT data from normal patients to use we preferred to find a group of patients without evidence of lung disease (group NL) directly collected in our hospital to minimize measure variability due to CT different techniques and observer evaluation in other centres.
Q2. Are there any common characteristics/features within the COVID-19 patients who died that are related to the increases in caliber (e.g. SOFA, APACHE, oxygenation, crackles, sqwuaks, etc).
REV2 Q2: We are sorry but we have no data about COVID-19-related death available for our patients. Anyhow, we don’t think the lack of these data really decreases the importance of our results that focus on pulmonary embolism not on death.
Q3.There are a lot of studies analysing CTs with deep learning techniques. The authors are urged to review the literature from the past 2 years for any studies that are directly or indirectly linked to the current paper.
REV2 Q3: We thank the reviewer for the suggestion and we added some articles recently published about CT analysis in COVID patients even with deep learning technique (please see page 4 at lines 81-84).
1) A deep learning algorithm using CT images to screen for Corona virus disease (COVID-19). Wang S, Kang B, Ma J, Zeng X, Xiao M, Guo J, Cai M, Yang J, Li Y, Meng X, Xu B. Eur Radiol. 2021 Aug;31(8):6096-6104. doi: 10.1007/s00330-021-07715-1. Epub 2021 Feb 24.
2) COVID-AL: The diagnosis of COVID-19 with deep active learning. Wu X, Chen C, Zhong M, Wang J, Shi J. Med Image Anal. 2021 Feb;68:101913. doi: 10.1016/j.media.2020.101913. Epub 2020 Nov 26.
3) Two-Stage Deep Learning Framework for Discrimination between COVID-19 and Community-Acquired Pneumonia from Chest CT scans. Abdel-Basset M, Hawash H, Moustafa N, Elkomy OM. Pattern Recognit Lett. 2021 Dec;152:311-319. doi: 10.1016/j.patrec.2021.10.027. Epub 2021 Oct 29.
4) A multi-center study of COVID-19 patient prognosis using deep learning-based CT image analysis and electronic health records. Gong K, Wu D, Arru CD, Homayounieh F, Neumark N, Guan J, Buch V, Kim K, Bizzo BC, Ren H, Tak WY, Park SY, Lee YR, Kang MK, Park JG, Carriero A, Saba L, Masjedi M, Talari H, Babaei R, Mobin HK, Ebrahimian S, Guo N, Digumarthy SR, Dayan I, Kalra MK, Li Q. Eur J Radiol. 2021 Jun;139:109583. doi: 10.1016/j.ejrad.2021.109583. Epub 2021 Feb 5.
5) Using Artificial Intelligence to Detect COVID-19 and Community-acquired Pneumonia Based on Pulmonary CT: Evaluation of the Diagnostic Accuracy. Li L, Qin L, Xu Z, Yin Y, Wang X, Kong B, Bai J, Lu Y, Fang Z, Song Q, Cao K, Liu D, Wang G, Xu Q, Fang X, Zhang S, Xia J, Xia J. Radiology. 2020 Aug;296(2):E65-E71. doi: 10.1148/radiol.2020200905. Epub 2020 Mar 19.
Q4. ARDS is very much related with COVID-19, but it is also present in other lung diseases such as COPD, asthma etc. How can ARDS be predicted from the proposed biomarker?
REV2 Q4. Thank you for the comment. The aim of our study was to provide distinctive radiologic features of COVID-19 pneumonia that might be useful as a diagnostic tool and alert radiologists to consider it when evaluating CT scans of patients at risk of pulmonary embolism. Although this is not a prognostic study, neither PaO2/FiO2 ratio nor ROX index (measured at the Emergency Department, at admission and, at patients’ clinical worsening) was associated with the V/B ratio of segmental or subsegmental branches as well as with the V/B ratio in the area of lung opacification (please see page 10, lines 216-219).
Round 2
Reviewer 1 Report
this article has been well revised.
Reviewer 2 Report
The authors have addressed the reviewers' comments.